# Indoor Radon Measurements in Finnish Daycare Centers and Schools—Enforcement of the Radiation Act

**DOI:** 10.3390/ijerph17082877

**Published:** 2020-04-21

**Authors:** Katja Kojo, Päivi Kurttio

**Affiliations:** Radiation and Nuclear Safety Authority (STUK), 00880 Helsinki, Finland; paivi.kurttio@stuk.fi

**Keywords:** radon, surveillance, school, daycare center

## Abstract

Background: Indoor radon exposure is the second leading cause of lung cancer. Finnish radiation legislation obligates employers to measure indoor radon concentrations in workplaces, including schools and daycare centers, if they are in radon prone areas. Surveillance campaigns were conducted to ensure that the required radon measurements were performed and to gain knowledge on current indoor radon levels in daycare centers and schools. Methods: Daycare centers located in the high-radon risk municipalities were identified. Schools where indoor radon level measurements were obligatory but not performed, were identified. Results: Indoor radon measurements were performed in 633 daycare centers where the mean radon concentration was 86 Bq/m^3^ and the median 40 Bq/m^3^. The radon level was greater than 300 Bq/m^3^ in 8% (*n* = 49) of daycare centers. The radon measurements were performed in 1176 schools, which is 95% of the schools to be measured. The mean radon concentration was 82 Bq/m^3^ and the median 41 Bq/m^3^. The radon levels were greater than 300 Bq/m^3^ in 14% (*n* = 169) of the schools. Conclusions: The systematic surveillance campaigns by the radiation protection authority were very efficient in order to ensure that the measurements are performed in schools and daycare centers. The campaigns also reduced the radon exposure of employees, children, and adolescents, where necessary.

## 1. Introduction

Indoor radon exposure is the second leading cause of lung cancer [1]. Currently it is estimated that slightly less than 300 new lung cancer cases occur yearly in Finland due to exposure to indoor radon. Lung cancer risk due to radon exposure accumulates the entire lifetime and therefore radon exposure in childhood may pose a significant long-term risk. Usually, most of the radon exposure occurs at homes and to a minor extent in other buildings, e.g., in workplaces. The average yearly work time of Finnish workers is approximately 1600 h [2]. Finnish radiation legislation obligates employers to measure indoor radon level of the workplace if there is a reason to suspect that the indoor radon level may exceed the national reference level for indoor radon concentration (300 Bq/m^3^). Measurements must be performed at least if the workplace is located (a) in an area where more than 10% of the indoor radon levels of the previously measured buildings exceeds the reference level, (b) on an esker or the ground underneath the workplace building is air-permeable, i.e., sand or gravel soil formations, (c) completely or partly underneath the ground level, or (d) the waterworks distributes household water derived from ground water. In case the workplace situates on the building’s second floor or above, measurements are not required.

In Finland, Radiation and Nuclear Safety Authority (STUK) is the responsible authority for regulatory control of radon in workplaces. Before the current Radiation Act (December 2018), the employer was not obliged to submit the results of the indoor radon measurements to STUK, if the concentration was smaller than 400 Bq/m^3^. Therefore, it was not known how many of the Finnish daycare centers and schools were measured and what the indoor radon levels were. It was anticipated that the responsibility to measure indoor radon level in workplaces was not well known and thus the required radon measurements were not performed in many of the schools and daycare centers. Surveillance campaigns addressed to daycare centers and schools were anticipated to be an efficient way to increase the radon awareness among employers.

The purpose of these two systematic surveillance campaigns was to increase the coverage of indoor radon measurements in daycare centers and schools and to reduce radon exposure of adults and children. The campaigns were conducted in order to gain knowledge on current indoor radon levels in Finnish daycare centers and schools. Indoor radon measurements in daycare centers and schools were also anticipated to increase the general awareness and public interest towards radon issues and encourage to make radon measurements at homes as well.

## 2. Materials and Methods 

### 2.1. Data Collection from Daycare Centers

The study was conducted as two separate projects, daycare centers and schools, with slightly different methods. We identified all the daycare centers located in Finnish municipalities where more than 10% of the indoor radon levels of the previously measured buildings exceeded the reference level (400 Bq/m^3^ at that time). There were altogether 61 of such municipalities at the beginning of the study (November 2014). We identified 945 daycare centers in these 61 municipalities. A contact letter (by e-mail) was sent to municipal directors of early childhood education (mainly in the case of public daycare centers) and/or to a director of a specific daycare center (mainly if the daycare center was operated by a private firm). Directors were asked to report whether the information on the daycare centers in the municipality was correct and whether the indoor radon was measured in the daycare centers. If the indoor radon was not measured, it was instructed to be measured. The costs of the radon measurements were paid by the operator or the municipality. The measurement results were submitted to STUK. In cases of outdated indoor radon concentrations (measurement performed more than 10 years ago), it was encouraged to repeat the measurements. Additionally, if major renovation work had been made in the daycare center after the previous indoor measurements, the measurements were instructed to be repeated. 

Those daycare centers that measured indoor radon levels using STUK’s radon detectors, were asked to report the number of employees in each measurement point. Information on number of employees was not asked from the daycare centers that used radon detectors provided by other than STUK. The number of children in the daycare center was inquired if the radon concentration turned out to be greater than 400 Bq/m^3^ in any measurement point. 

### 2.2. Data Collection from Schools

A contact letter (e-mail) was sent to the Environmental Health Authority office in every Finnish municipality in September 2016. Municipal Health Protection sector was the best contact in this project since the Health Protection Authorities monitor the indoor radon levels in schools according to Finnish Health Protection Law. In the contact letter we asked for information on all schools in the municipality where indoor radon level measurements were obligatory but had not yet been performed. Information was asked on elementary schools, special schools, high schools, trade schools, and special trade schools. Later in this article, we cite all these as “schools”. In the contact letter, it was instructed to measure the indoor radon level in these schools and submit the results to STUK. The costs of the radon measurements were paid by the operator or the municipality. The number of employees as well as the number of children and/or adolescents studying in these schools was inquired. They were also instructed to inform if there were no such schools in the municipality where radon was obliged to be measured. If previous indoor radon measurement results were available, the measurements were instructed to be repeated in case the previous measurements were outdated or major renovation work was made in the school building after the previous measurements.

### 2.3. Indoor Radon Measurements

At least two-month radon measurements in the daycare centers and schools were carried out using three types of radon detectors: “Radonpurkki” by STUK (≈73% of all measurements), AlphaRadon detector (≈5%), and Radtrak^2^ by Radonova (≈22%). All detectors comply with the requirements set by the Guide ST 1.9 [3] and have been approved for radon measurements in workplaces and homes by STUK. The detector by STUK is accredited for radon measurements in indoor air by FINAS Finnish Accreditation Service (laboratory code T167). STUK’s passive alpha track detectors employs Makrofol® film (Covestro, Dormagen, Germany). After the exposure, the films are etched electrochemically and counted automatically using an autosampler, microscope, and pattern recognition software. The minimum and maximum detectable radon concentrations in a 60-day sampling are 13 and 14,000 Bq/m^3^, respectively.

For the STUK’s “Radonpurkki” detector, the uncertainty level from autumn 2016 onwards was calculated as result-specific with k = 2, which is equivalent to approximately 95% confidence level. Before autumn 2016, the uncertainty level was +/-20% with k = 2. For Alpha Radon and Radonova detectors, uncertainty levels are result-specific. In this article, the results are reported without the uncertainty level.

The duty of the employer to submit the radon measurement results to STUK is defined in the Radiation Act. The measurement period lasted from the beginning of November until the end of April. The passive detector measurement result is the average radon concentration of the entire measurement time. 

Daycare centers and schools were instructed to measure with enough detectors. Every individual building was to be measured at least with one detector. In addition, STUK instructed that there should be one detector for every 200 m^2^ in the building. If the daycare center was situated in an apartment, the workplace was instructed to measure with two detectors. 

Depending on the size and the number of separate buildings in the daycare center or school, the number of radon measurement detectors varied from 1 to 18 in daycare centers and 1 to 40 in schools. Altogether 430 daycare centers and 964 schools measured indoor radon with more than one radon measurement detector. The indoor radon level of these daycare center and school buildings was calculated as the average of the measurement results. If a daycare center or school measured the indoor radon levels in only one measurement point (i.e., with one detector), that result was considered as the radon level of the entire daycare center or school.

The radon statistics described in this article are based on actual measurement results, i.e., seasonal correction factor was not used for the measured radon concentration. Before the current Radiation Act (December 2018), STUK imposed corrective actions based on the actual measurement results. After the enforcement of the current Radiation Act, the radon concentration measured during the measurement period was multiplied with 0.9 (a seasonal correction factor) and that result was compared to the reference level of radon at workplaces.

### 2.4. Data Analysis

Descriptive statistics were determined, that is, the arithmetic and geometric mean, median, and maximum values of radon concentration. Additionally, the proportion of values greater than radon concentrations of 200, 300, or 400 Bq/m^3^ were determined. STATA-15 software (StataCorp, College Station, Texas, United States) was used in all analyses [4].

## 3. Results

### 3.1. Daycare Centers

Indoor radon measurements were performed altogether in 633 daycare centers. This was 67% of the daycare centers (*n* = 945) that were contacted. Some daycare centers did not have to perform the measurements, for example, if the daycare center was on the second floor (or higher) of the building. Additionally, those daycare centers that reported that they will close soon did not have to perform radon measurements. Table 1 summarizes the status of the daycare centers. There were 862 daycare centers in the municipalities where radon measurements in workplaces are mandatory, but only 39% (*n* = 335) of those daycare centers had performed the radon measurements before this surveillance campaign. 

During the first measurement period (November 2014–April 2015) after the contact letter was sent, 482 daycare centers performed the radon measurements. A reminder letter was sent to those daycare centers that did not perform the measurements and they were reminded to perform them during the next measurement period. However, for some daycare centers one reminder was not enough and the last radon measurements were performed during the measurement period of 2017–2018.

The greatest radon level measured in a daycare center building was 2426 Bq/m^3^ (Table 2). This was also the greatest single measurement result since the daycare center in question was measured only in one measurement point. Radon concentration varied in some cases within a daycare center building. For example, in one daycare center, where 12 detectors were used, the lowest measured radon concentration was 72 Bq/m^3^ and the highest 1460 Bq/m^3^. In another daycare center with six detectors used, the radon concentrations were 41 and 702 Bq/m^3^, respectively. Indoor radon level was greater than 200 Bq/m^3^ at least in one measurement point in 14% (*n* = 86) of daycare centers, greater than 300 Bq/m^3^ in 8% (*n* = 49), and greater than 400 Bq/m^3^ in 4% (*n* = 26) daycare centers.

If the indoor radon level in a daycare center in any of the measurement points was greater than the earlier reference value 400 Bq/m^3^, STUK imposes corrective actions, for example, to make additional measurements with a continuously measuring radon detector or to reduce the indoor radon level with a suitable radon remediation method. After the daycare center radon campaign, a new Radiation Act with a new reference level of 300 Bq/m^3^ has come into force in Finland. Those daycare centers where the radon concentration was less than 400 Bq/m^3^ but greater than 300 Bq/m^3^ are subject to law transition time regulations. These daycare centers must take actions to reduce the indoor radon level within ten years. 

If a daycare center performed radon remediation action, its successfulness had to be confirmed with a radon measurement of at least two months. The results of these re-measurements are obliged to be sent to STUK. STUK enforced that all the necessary requirements to reduce indoor radon exposure were fulfilled in the daycare centers. At this point (March 2020) all the daycare centers that received regulatory orders have fulfilled the requirements. STUK has received the re-measurement results from all these daycare centers and them, the radon concentrations have been smaller than the reference level in all of them. All the measurement results were published on STUK’s website. 

### 3.2. Schools

Out of 313 municipalities in Finland which received STUK’s contact letter, 192 municipalities have provided measurement results at least from one school by March 2020. Ten municipalities replied to the contact letter that there are schools that should be measured but the measurement results have not yet been delivered to STUK by March 2020. STUK has sent a reminder letter (e-mail) to these 10 municipalities that are still lacking the radon measurements. 

A total of 1242 schools with a radon measurement obligation were reported to STUK (Table 3). During the first measurement period (November 2016–April 2017), 852 schools performed radon measurements and by March 2020, 1176 (95%) schools had performed the measurements. There are still 56 schools that have not performed the radon measurements.

The greatest radon level measured in a school building was 4205 Bq/m^3^ (Table 2). The highest single concentration was 14,386 Bq/m^3^. Similar to the daycare centers, radon concentration varied considerably within one school building in certain cases. For example, in the school where 11 detectors were placed to various measurement points, the lowest radon concentration was below the detection limit (i.e., <20 Bq/m^3^) and the highest 1400 Bq/m^3^. Indoor radon level was greater than 200 Bq/m^3^ at least in one measurement point in 22% (*n* = 264) of schools, greater than 300 Bq/m^3^ in 14% (*n* = 169), and greater than 400 Bq/m^3^ in 10% (*n* = 121) of schools. The distribution of radon concentrations is very similar to daycare centers; it is clearly skewed, i.e., the majority of the results are in the lowest categories.

As with daycare centers, STUK imposes corrective actions to be taken, i.e., additional measurements or to reduce the indoor radon level with some suitable radon remediation method (and confirm the successfulness of the corrective actions with re-measurements) in schools where the indoor radon level in any of the measurement points was greater than the earlier reference value 400 Bq/m^3^. For those schools that provided the measurement results to STUK after the new Finnish radiation legislation came in action (i.e., December 2018 onwards), the reference level of 300 Bq/m^3^ was applied. STUK enforces that all the necessary requirements to reduce indoor radon exposure are to be done in schools. 

### 3.3. Number of Employees, Children, and Students Affected by These Campaigns

The number of employees in daycare centers was not reported systematically. Part of the daycare centers reported the number of employees for each measurement point, as instructed, but certain daycare centers reported the total number of the employees for all the measurement points. We used the given information to estimate the total number of employees reported by 368 daycare centers. The estimated total number of people working in these daycare centers was 2000. The number of children was inquired from those daycare centers where the radon concentration in any of the measurement points exceeded 400 Bq/m^3^. This number was 599.

A total of 759 schools reported the number of employees and 857 schools the number of students. The total number of people working in these schools was 19,553 and the total number of students in these schools was 195,779. Out of those schools (*n* = 169) where the indoor radon level was higher than 300 Bq/m^3^ only 48 schools reported the number of employees and 57 schools the number of students. In these schools, the total number of students and employees was 1361 and 11,302, respectively.

## 4. Discussion

### 4.1. Main Findings of the Study

Radon measurements have been conducted in Finland since the 1980s. Despite the long history of radon measurements, there were many unmeasured daycare centers and school buildings before these radon campaigns, but as a result of the campaigns, the required measurements were performed. The results of the campaigns indicate that 8% of daycare centers and 14% of schools in Finland had greater radon concentration than the current reference level, 300 Bq/m^3^. All the daycare centers have successfully reduced the radon exposure in their premises. Regulatory control authority, STUK, assures that in schools where radon measurements and exposure reduction are lacking, these measures are performed.

It is not possible to estimate the exact number of people whose radon exposure was reduced because of the inadequate reporting of the number of people. However, as a result of these campaigns, the radon exposure of many employees, children, and adolescents was reduced. Furthermore, it was not possible to find out how many people spend time, and for how long, in those parts of the buildings with high radon concentrations. 

Radon measurements are mandatory in certain schools and daycare centers in Finland. However, many of them had not measured radon until these campaigns. Only approximately 40% of those daycare centers, where radon measurements in workplaces are mandatory, had performed the measurements earlier. There might be several reasons for that. Employers of daycare centers and schools are simply not aware of their statutory responsibility and, thus, there is a clear need to increase the information to employers. Additionally, the reluctancy of the employers towards radon measurements may be due to the fact that the employer is responsible for radon measurements and thus also for the costs of the measurement and the possible radon remediation actions. The law enforcement process is still ongoing for some of the schools. Clearly, certain responsible parties of the schools and daycare centers have not understood that there are obligations in the Radiation Act for them.

### 4.2. Strengths and Weaknesses of the Study

The current study shows a unique setting, where radon campaigns are conducted by the national radiation safety authority. We did not find publications form recent years from similar campaigns. National authorities in Sweden and Norway have measured indoor radon levels in schools, preschools and daycare centers. The results of these measurement have not been published, but there are protocols written for these actions; in Sweden in Swedish [5] and in Norway in English [6].

Radon measurements, in practice, were carried out by the daycare centers and schools. Thus, STUK was not able to ensure that all the given instructions, i.e., positioning of radon detectors correctly into the measurement points, were followed. Daycare centers and schools were instructed to measure with a sufficient number of detectors. Using enough detectors is important, since radon concentration might vary considerably between measurement points. However, STUK was not able to control that enough detectors were used. A great number of schools and daycare centers were measured during this study. In many previous studies (Table 4), the number of school and daycare centers measured was not as extensive as in the current study. This study covers all the schools and the majority of the daycare centers in Finland where radon measurements are mandatory. 

### 4.3. Comparison to Radon Levels in Previous Studies

Several studies have been published in the recent years regarding indoor radon levels in schools and daycare centers. We found 18 publications in English in scientific journals during the last years (year 2014 onwards) that reported the average radon levels. In those studies, the terminology of schools and daycare centers might differ from this study. “Nursery”, “kindergarten”, and “preschool” in the above-mentioned studies are comparable to “daycare center” in this study and, respectively, “primary school” to “school”. Table 4 summarizes the main characteristics of these studies and the reported mean (arithmetic and/or geometric) radon indoor concentrations. 

In our study, the average radon (AM) concentration was 86 Bq/m^3^ in daycare centers and 82 Bq/m^3^ in schools, the geometric means were 39 and 41 Bq/m^3^, respectively. Compared to recent studies, the Finnish radon levels in schools and daycare centers are smaller than 11 out of 18 average (one median) radon levels reported in previous studies (Table 4). This result is quite surprising since regarding radon levels worldwide, the indoor radon levels in Finland have been among the highest [25]. The Finnish study was conducted in those schools and daycare centers where indoor radon measurements are mandatory, i.e., in high radon risk areas. Only some of the previous studies were targeted to schools and daycare centers in high radon risk areas, so this does not explain the higher radon levels in previous studies compared to this study. It is possible that the Finnish daycare centers and schools are built in a very radon-safe way, although this study offers no data to investigate that hypothesis.

In general, radon levels in workplaces and buildings with public access are lower than radon levels in homes in Finland. The mean radon concentration in Finnish homes is 96 Bq/m^3^ [26], which is somewhat higher than the mean radon level found in daycare centers and schools in this study.

## 5. Conclusions

The systematic surveillance campaigns of the radiation safety authority are efficient to ensure that radon measurements are performed in those schools and daycare centers where the measurements are obligatory. STUK’s involvement in assuring that remediation is done where needed, resulted in reduction of radon exposure concerning many employees, children, and adolescents.

In Finland there is a good protocol for indoor radon measurements in schools, daycare centers, and other workplaces; the Radiation Act obligates the employer to perform radon measurements in radon prone areas. STUK will continue the surveillance campaigns applying the graded approach on indoor radon concentration in various workplaces.

## Figures and Tables

**Table 1 ijerph-17-02877-t001:** Status of the daycare centers reported to Radiation and Nuclear Safety Authority (STUK) and whether the indoor radon measurements were performed in the campaign.

Status of the Daycare Center Reported to STUK	Radon Measurements in the Surveillance Campaign		Total
	Yes	No	
Radon measurements will be performed	488	-	488
Radon measurements not necessary	9	74	83
Indoor radon measured earlier	97	238	335
No reporting to STUK	39	-	39
Total	633	312	945

**Table 2 ijerph-17-02877-t002:** Main descriptive statistics of indoor radon measured in daycare centers and schools in the campaigns.

	Daycare Centers	Schools
Arithmetic mean (Bq/m^3^)	86	82
Geometric mean (Bq/m^3^)	39	42
Median (Bq/m^3^)	40	41
Maximum (Bq/m^3^)	2426	4205
% greater than 300 Bq/m^3^	8	14

**Table 3 ijerph-17-02877-t003:** Status of the schools with radon measurement obligation reported to STUK and whether the indoor radon measurements were carried out by March 2020.

Status	Count
Radon measurements	1176
No radon measurements because school was shut down	10
Radon measurements not performed	56
Total	1242

**Table 4 ijerph-17-02877-t004:** Summary of the main results in previous studies conducted in recent years in schools or daycare centers regarding indoor radon measurements. AM = Arithmetic mean, GM = Geometric mean.

Country	Type of Study	Average (AM) Radon Level Bq/m^3^	Average (GM) Radon Level Bq/m^3^
Finland (this study)	Daycare centers and schools	82 (Schools), 86 (Daycare centers)	42 (Schools), 39 (Daycare centers)
Portugal (Porto and Bragança district) [7]	Nurseries, primary, and preschools	62 (Porto), 193 (Bragança)	47 (Porto), 147 (Bragança)
Turkey [8]	Primary schools	49	42
Serbia [9]	Primary schools	119	-
Bulgaria [10]	Kindergartens	132	101
Republic of Macedonia [11]	Schools	88	76
Czech Republic [12]	Schools	204 (reconstruction), 149 (non-reconstruction)	-
Hungary, Poland and Slovakia [13]	Kindergartens	233 (Hungary), 90 (Poland), 317 (Slovakia)	231 (Hungary), 78 (Poland), 214 (Slovakia)
Russia [14]	Kindergartens	59	42
South Italy [15]	Schools	215	-
Italy [16]	Schools	-	92 (median only reported)
Korea [17]	Schools	97 (spring semester), 158 (autumn semester)	-
Bosnia and Herzegovina [18]	Schools	128	99
Kosovo [19]	Schools	198	-
Saudi Arabia (Riyadh) [20]	Schools	17	17
Serbia (Kragujevac city) [21]	Schools and kindergartens	60	-
Sudan [22]	Schools	59	-
Switzerland [23]	Schools	158	-
Portugal [24]	Schools	197	197

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
