# Peer review of "Indoor Radon Measurements in Finnish Daycare Centers and Schools—Enforcement of the Radiation Act"

_ijerph, 2020, doi:10.3390/ijerph17082877_

Round 1
Reviewer 1 Report
Nowadays the „radon-problem” is a very important question from the aspect of public health. The identification of the radon-prone areas and building with high indoor radon concentration is a very important task.
In this study, indoor radon measurement was performed in 633 daycare centres and 1176 schools in Finland. The measurements were carried out by passive alpha track detectors. STATA-15 software was used for the analysis of data.
This manuscript is a well-written, coherent and clear. The description of the survey is correct. The analysis of the data is adequate. I have only soma question or comments.
In Section 2.3.:
a) Which type of track detector was used in the survey?
b) The measurement period was from November until April, and the exposure time was at least 2 month. It means, there were some buildings, where the measurement was in wintertime and others in the early springtime.
Did you do any seasonal correction with the data? (Or in Finland the seasonal variation in this period is negligible?)
In Section 3.1. and 3.2.
a) Do you have any information about the efficiency of the radon remediation method in building with the highest radon level?
b) Did you find any reason for the very high (4 205 and 2 426 Bq/m3) indoor radon concentration? Are these building any special areas?
In Section 4.3.
Did you compare the results from this survey with the results from previous dwelling surveys in Finland? Is there any similarity between them?
Reviewer 2 Report
The submitted manuscript deals with systematic surveillance campaigns conducted by STUK to assess the indoor radon level in Finnish daycare centers and schools located in radon prone areas. Depending on the radon level, mitigation measures were undertaken in order to reduce the people exposure.
The research has been clearly described in the paper, and the obtained results as well.
I recommend the publication of this important survey with a few minor revisions (or just suggestions) that follow:
L.10: in workplaces
L.11: daycare centers,
L.22: The campaigns (were also aimed at) reducing … children and adolescents, where necessary.
L.61: at the beginning
L.141: with a
L.195-196: Please rephrase "Regulatory control authority, STUK, follow that in schools the lacking"; maybe as "Regulatory control authority, STUK, assures that, in schools, lacking radon measurements and exposure reduction are performed/carried out"
L.198: because of the inadequate/incomplete reporting of the number of people.
L.200: spend time, and for how long,
L.219: Radon measurements, in practice, were carried out ...
L.230: In those studies,
L.232: “daycare center” in this study and, respectively, “primary-school” to “school”
L.252-253: A suggestion: "are obligatory. As a result, the radon exposure ..." could be rephrased as : "are obligatory. STUK involvement in assuring that remediation be done where needed, resulted in radon exposure reduction of many employees, children and adolescents.
Finally, regarding further surveys, the building materials role could be analysed, if possible, since it could influence the radon exposure of people living or working at higher floors also
Reviewer 3 Report
The manuscript reports valuable information, but it is organized more as an administrative report than as a research paper. Major revision is recommended that has to address important points, in particular:
- The detectors used are not adequately described. Their type and characteristics should be given either in the text or by a reference.
- The authors refer to the "reference level" of 300 Bq/m3. Upon my knowledge in EU the reference level recommended to the member states (e.g. Finland) should be based on the annual average radon activity concentrations. If this is the case in Finland, this should be clearly written. If something else - it should be also explained. If the reference level is the annual average, the authors should discuss whether they have applied seasonal correction factors given the exposure period covers only part of the year. Was the exposure time one and the same for all measurements?
- Some terms used by authors don't sound well for me: e.g. I suggest to replace "the total indoor radon level" with "mean for the building radon level", or "mean for the center/school radon level".
- The results are given without uncertainty info (e.g. 2426 Bq/m3 - what is the uncertainty?).
- The reader would benefit if the authors include some info about the variations between the results in different points within one center/school.
- As the authors say in the Conclusion that "As a result the radon exposure ....... was reduced" they should provide more info in the text which factor of reduction was achieved, info about radon "before and after" etc. in order to justify such important conclusion.
If/when the authors succeed in the revision recommended, this work can be reconsidered.
Round 2
Reviewer 3 Report
Overall, the raised points have been addressed satisfactory.